# Safety Risk Identification Method for Railway Construction in Complex and Dangerous Areas

Peng Wang [1], Qiang Wei [2], Guotang Zhao [1,2], Jingchun Wang [3] and Yang Yin [4,*]

1   School of Civil Engineering, Beijing Jiaotong University, Beijing 100044, China
2   China State Railway Group Co., Ltd., Beijing 100844, China
3   School of Safety Engineering & Emergency Management, Shijiazhuang Tiedao University, Shijiazhuang 050043, China
4   School of Civil Engineering, Central South University, Changsha 410017, China
*   Correspondence: yinyang@csu.edu.cn

**Abstract:** Safety risk identification is the premise and foundation of safety risk management for railway construction. However, due to some characteristics of railway projects, which include large volumes of work, complex construction environments, and long construction cycles, etc., the risk factors of railway projects are often hidden in all stages of engineering construction. It results in the comprehensive identification of safety risks of railway projects being usually difficult, and this problem is more serious when the railway is constructed in complex and dangerous areas. Therefore, to identify the safety risks comprehensively, this paper constructs a safety risk identification method applicable to railway construction in complex and dangerous areas. This method studies the spatial and temporal distribution of risks and their relationship with subprojects by using a work breakdown structure (WBS), a risk breakdown structure (RBS), grid-based management, and forming a safety risk identification matrix, which can help researchers analyze the characteristics of risks. In order to verify the effectiveness of the method, the A Railway, which is located in the western of China, was selected as a case study, and risk identification for its civil engineering was carried out. The research results show that in the construction process of the A Railway, the main types of safety risks suffered by various branch projects were different. In addition, some risk factors only appeared at specific times in space, and there is a strong interaction between these risk factors. Based on this method, safety risk identification can intuitively discover the spatial and temporal distribution of risk factors and analyze the interaction between risk factors, which can provide help for the formulation of targeted risk control measures.

**Keywords:** railway construction; risk identification; WBS; RBS; grid-based management

## 1. Introduction

Safety is the eternal theme of human life and production activities [1]. In engineering construction, there are very complex interactions among these factors: construction personnel, construction organization, construction equipment, and construction environment, and the uncoordinated interactions among these factors lead to the generation of safety risks [2]. As a major engineering project, railway projects are characterized by complex construction technology, variable construction environments, and difficult for safety risk control [3], which results in various types of unavoidable safety risks during the construction process. In this paper, complex and dangerous areas refer to the western region of China, which is characterized by complex geological conditions, a harsh climatic environment, high posters, frequent earthquakes, and so on [4]. The "complex" refers to this area because there exist multiple safety risk factors, such as geological disasters, natural disasters, and extreme climate, and these factors usually interact with each other, resulting in a complex construction environment. The "dangerous" means that the safety risks of railway construction in this area are usually serious, which will lead to serious risk events if not controlled

in time. In complex and dangerous areas, railway construction safety risks are difficult to fully analyze and understand [5], and their uncertainty, suddenness, complexity, and high destructiveness are prominent. The complex and dangerous areas not only affect the operation of the railway system [6,7], but also pose challenges for the construction of the rail infrastructure. Frequent landslides, mudslides, earthquakes, and other natural disasters have posed a great threat to the safety of the construction personnel and the railway. In addition, rockburst, fault zones, high ground temperatures, and other adverse geological problems bring great troubles to the construction of tunnel engineering. In the construction process of railway projects in complex and dangerous areas, more comprehensive safety measures have to be taken to ensure that there will be no major safety accidents in the construction process. Enhancing the level of safety risk identification and control capabilities and constructing a compatible railway construction safety risk control system [8] are important means to reduce the incidence and losses of risky accidents.

Risk identification refers to using appropriate methods to identify the risk factors in the implementation of a project and preparing risk identification reports. Safety risk identification is the premise and foundation of safety risk assessment and control, and comprehensive, accurate, and effective risk identification is of great significance to the formulation of construction plans and risk control measures [9]. Many scholars have conducted in-depth research in the field of construction project safety risk identification and formed rich research results. Goh and Chua [10] introduced a case-based reasoning (CBR) approach to the risk identification process for building safety. Based on fuzzy hierarchical analysis, Khademi et al. [11] identified and analyzed the safety risks that may be suffered during TBM construction to provide a basis for selecting the appropriate one. By analyzing the common model of risk assessment in the Slovak railway system, Leitner [12] pointed out that identifying risks needs clarification of the definition and composition of risks, analyzed the causes and development sequences of risk events, and proposed five steps for risk identification and the development of risk events. Based on the unknown and uncontrollable nature of safety risks, Vishwas et al. [13] pointed out that risk identification is one of the important steps in construction safety risk management and can provide an important research basis for risk analysis. In response to the lack of systematicity and comprehensiveness of traditional methods, Liu et al. [14] proposed a root-state hazard identification method. Li et al. [15] proposed a new accident–cause network to identify railway risks and prevent corresponding accidents. The above research results point out that safety risk identification in the construction process plays a vital role in safety risk control, and a comprehensive and accurate safety risk identification can help managers formulate more targeted risk control measures and improve the efficiency and level of risk control.

With the advent of the industry 4.0 era [16], safety risk identification is becoming more and more closely linked to emerging technologies. Ding et al. [17] proposed a subway construction safety risk identification system that incorporated four identification algorithms. The system identifies construction drawings, which allows for easy and fast access to engineering data, and obtains its relationship with safety risks in the knowledge base. Shi et al. [18] employed topic mining techniques and data flow algorithms for risk identification and safety management, demonstrating the potential of large unstructured data. Aliyari et al. [19] carried out a preliminary risk analysis of a bridge with the help of UAVs. Ji et al. [20] constructed the safety risk evaluation system of bridge engineering based on the WBS–RBS analysis method. Zhang et al. [21] constructed the safety risk evaluation system of deep foundation pit projects based on the WBS–RBS analysis method, also. However, existing studies mostly focus on the identification of risk factors from a static perspective, ignoring the dynamic change process of risk factors, which will lead to the inability of researchers to comprehensively understand safety risks.

Railway construction safety risk identification [22] is mainly conducted using suitable models and methods to find out the risk factors affecting railway construction safety and studies the causes and categories of risk factors. For railway construction in complex

and dangerous areas (RCCDA), risk factors are often hidden in all stages of engineering construction [23]. Due to the characteristics of RCCDA, such as large volumes of work, complex construction environments, and long construction cycles, its risk factors usually interact and are obviously distributed at specific times or places. The interaction of risk factors refers to multiple risk factors that interplay when they appear at the same time or place. The safety risks which are caused by the interaction of risk factors are difficult to control through simple control measures. The distribution of risk factors at a specific time or place means that some risk factors only appear at a specific time or place. The reason for this phenomenon is that complex and dangerous environments have great influences on railway engineering construction, such as the risks of fault zones and rockburst, which are produced by geological factors, and the risks of strong winds and snow, which are produced by the weather factors. To control such risks, it is necessary to analyze the characteristics of risk factors in detail and take targeted control measures. So, the risk identification work must be combined with the characteristics of risks with the help of appropriate technical methods and tools to improve the efficiency of risk identification and to avoid missing significant risk sources [24].

Therefore, according to the characteristics of RCCDA and the requirements of railway construction safety risk identification, this paper constructed a new safety risk identification method based on the grid–time–work breakdown structure–risk breakdown structure (G–T–WBS–RBS) matrix by integrating the methods of WBS, RBS, and risk management grid, etc. This risk identification method can help researchers quickly discover the spatial and temporal distribution and interaction of safety risk factors by constructing a G–T–WBS–RBS matrix. Based on the spatial and temporal distribution and interaction of risk factors, researchers can analyze the causes of safety risks so as to formulate more targeted risk control measures according to the characteristics of safety risks.

## 2. The Safety Risk Identification Method Based on the G–T–WBS–RBS Matrix

Railway construction is characterized by technical complexity, large scale, and susceptibility to force majeure factors and accidents. Especially for RCCDA, the corresponding risk status changes dynamically with time and space [25]. Conventional identification methods have limitations for the safety risk identification of RCCDA, and it is difficult to accurately identify the spatial and temporal location of the risk occurrence and the real-time change state of the risk.

The WBS–RBS (work breakdown structure–risk breakdown structure) can systematically organize the risk patterns of projects [20], which facilitates risk planning response, data processing, and experience accumulation. However, at the same time, it cannot be well integrated with temporal and spatial changes. Therefore, the G–T–WBS–RBS (grid–time–WBS–RBS) risk identification method is introduced in this paper, and its general process is shown below:

1. Use the WBS to analyze railway projects and form the corresponding WBS tree. Based on the characteristics of the construction activities, form the railway construction safety risk identification grid.
2. Find the temporal location relationship between the WBS trees and the railway construction safety risk identification grid and build the G–T–WBS (grid–time–WBS) matrix.
3. Identify and sort the risks and form the RBS trees, then cross the G–T–WBS matrix and the RBS tree to form the G–T–WBS–RBS matrix.
4. Identify the construction activities and associated risks in each grid, then analyze the spatial and temporal distribution characteristics of the safety risks of RCCDA.

### 2.1. Safety Risk Management Grid of Railway Construction

The concept of the grid is derived from the Internet, an important information technology that has emerged internationally. As an emerging technology built based on network, grid is widely used in information management in many fields such as biomedicine [26],

ocean management [27], and railway transportation [28]. Grid-based management is based on the concept of Internet grid management, which divides the management objectives or objects into several grid cells according to certain criteria. At the same time, information technology is used to enable information exchange and resource sharing among the grid cells to achieve the purpose of improving management efficiency [29]. Compared with the traditional linear management model, grid management can achieve more accurate, more efficient, and more detailed management effectiveness.

Railway engineering is a typical linear project which is continuous and repetitive in the horizontal direction, and most of the processes and subprojects are also continuous and repetitive in the construction process. Accordingly, combined with the construction characteristics of linear engineering projects, a two-dimensional right-angle coordinate system, i.e., the time–location (T–L coordinate), can be used to describe the construction process or progress level of the railway. In the T–L coordinate, the spatial location of the work point (route mileage) is represented by the X-axis, and the duration time (project progress) is represented by the Y-axis. Therefore, the construction state of any work point in railway construction can be expressed in T–L coordinates according to its time–space.

T–L coordinates show the time, mileage, and constraints of railway construction activities, with a very strong visualization. According to the characteristics of railway construction, the process of railway engineering can be divided into three types: linear activities, bar activities, and block activities.

As shown in Figure 1, activities A, C, and E are linear activities (track laying, girder erection, etc.), activity B is a bar activity (special structures such as continuous beams), and activity D is a block activity (roadbeds, tunnels, etc.).

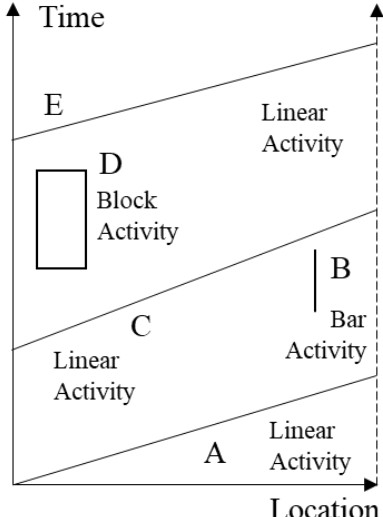

**Figure 1.** The T–L coordinate and activity types.

The spatial position of linear activities on the horizontal axis and the time on the vertical axis are mutually asymptotic. The linear activity has a certain linear relationship in spatial distance and time, and the slope of the straight line in the T–L coordinate reflects, to some extent, the rate (beam erection or track laying ergonomics) when executing the activity. The spatial position of the horizontal axis of the bar activity does not vary with the time of the vertical axis, and its construction duration is reflected by the Y-axis. The block activity is similar to bar activities. It spans a segment of the project mileage, while bar activities are performed at a fixed spatial location in the project.

Combined with the above research, in order to give full play to the role of grid management in the safety risk management of railway construction, it is divided as follows:

1.  Mileage length division of the railway construction safety risk identification grid (RCSRIG). According to the characteristics of railways, such as long lines, many points,

wide areas, and clear land boundaries, the RCSRIG should be divided according to the length of the mileage. According to the characteristics of railway construction and previous studies [30], the RCSRIG uses a line section of 200 m as the basic mileage length unit.

2. Time length division of the RCSRIG. Railway construction is a continuous activity. Once the project starts, no special circumstances will interrupt the construction process; construction safety risks have been accompanied by the entire process. In addition, the occurrence of risk has irregularity and suddenness. Therefore, the management of construction safety risks is equally continuous and lasting. At the same time, railway construction can be divided into stages such as construction preparation, main construction section, completion, and acceptance. Each stage has a certain time zone, corresponding to different safety risks. In addition, the project schedule is prepared in accordance with the days, and the actual construction is also in accordance with the daily plan. So, the RCSRIG uses a day as the basic time length unit.

## 2.2. Railway Construction WBS and RBS

The WBS consists of three elements: work, breakdown, and structure. Work indicates the completed work objectives, breakdown indicates the subdivision of work objectives into hierarchical structures, and structure indicates the establishment of an independent and interrelated hierarchy of each layer according to the system principle with a work objective as the center. The WBS is an engineering management tool developed on the theoretical basis of cybernetics, information theory, and system engineering. It is widely used in engineering activities such as schedule planning, resource planning, cost budgeting, personnel planning, and risk analysis.

The RBS is similar to the WBS, which is a breakdown structure with a work objective risk as the main body and is constructed with reference to the WBS. The safety risks of RCCDA are specific in terms of personnel, equipment, environment, and management. Personnel risk mainly considers the professional skill level and physical condition of personnel. Machinery risk mainly considers the operation and maintenance condition of machinery. Management risk mainly considers the management ability of relevant departments. Environmental risk mainly considers hydrogeological and climatic conditions [31].

## 2.3. The Safety Risk Identification Method Based on G–T–WBS–RBS

The grid management of the WBS can be realized by studying the position of the smallest unit of the WBS in the T–L coordinate based on its temporal and spatial properties. For example, if the railway construction is decomposed step by step, according to the time and space attributes of unit works, division works, and subprojects corresponding to the T–L coordinate, it is very clear to see when and where a certain construction activity is carried out.

The railway construction safety risk identification grid is an important prerequisite for G–T–WBS matrix construction. The RCSRIG is constructed based on the T–L coordinate, and the horizontal axis (X) of it indicates the mileage position, and the vertical axis (Y) indicates the time progress. The units of the RCSRIG are obtained by cutting the horizontal and vertical axes of the T–L coordinate according to some rules, and each unit represents a spatial–temporal segment of a railway line. Finally, the grid units are encoded and denoted by $G_{i,j}$, with $i$ as the horizontal coordinate and $j$ as the vertical coordinate. For example, 4 and 3 in the red circle in Figure 2 indicate that the abscissa of the unit is 4 and the ordinate is 3, so it will be marked as $G_{4,3}$ when it is encoded. Based on the previous study, the unit lengths of the horizontal and vertical coordinates are taken as 200 m and 1 day, respectively [32].

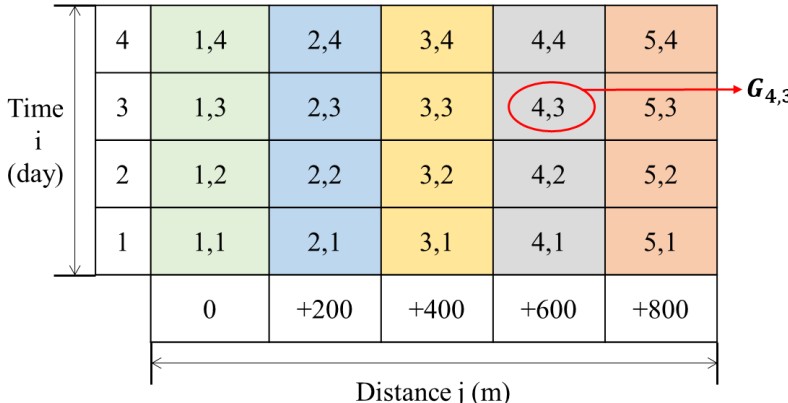

**Figure 2.** Railway construction safety risk identification grid.

For any construction activity, it can be divided into a linear activity, a bar activity, and a block activity according to the characteristics of it, and then expressed in the RCSRIG according to its temporal and spatial attributes, forming the railway construction schedule map as shown in Figure 3.

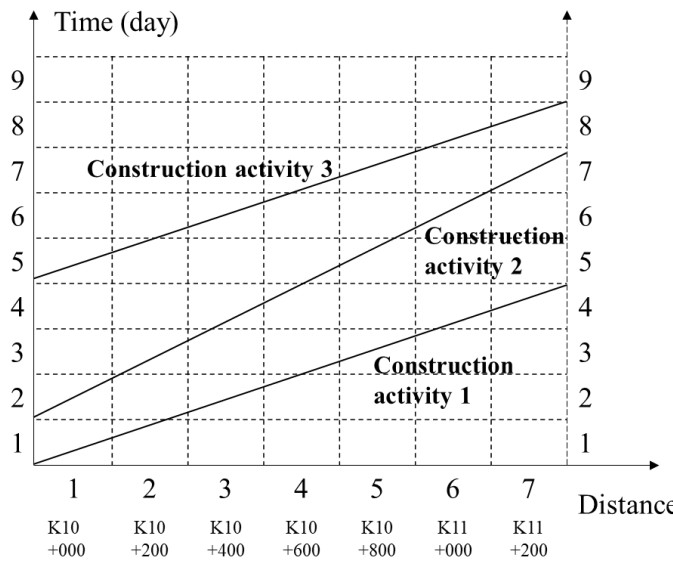

**Figure 3.** Railway construction schedule map based on the RCSRIG.

The railway construction schedule map reflects the spatial and temporal distribution characteristics of the construction activities. By analyzing the railway construction schedule map, the G–T–WBS matrix can be formed by assigning "1" to the units in which construction activities exist and "0" to the units which lack construction activities. Taking construction activity 1 in Figure 3 as an example, the G–T–WBS matrix shown in Figure 4 can be formed by organizing its construction schedule map.

The G–T–WBS matrix reflects the spatial and temporal attributes of the construction activity, and the RBS identifies and sorts the safety risks of railway construction layer by layer, comprehensively analyzing the risk factors existing in the process of railway construction. Based on the above analysis, the G–T–WBS–RBS matrix can be constructed by exploring the interrelationship between the spatial and temporal attributes of construction activities and risk factors. Firstly, according to the G–T–WBS matrix, only the "1" unit is retained. Secondly, the risk factors are identified and sorted in layers according to the sources. Finally, the GTWR matrix is formed by analyzing the safety risks that construction activities may encounter while working within each grid cell and organizing them. Taking construction activity 1 again as an example, the G–T–WBS–RBS matrix shown in Table 1

can be obtained by analyzing the interrelationship between its G–T–WBS matrix and risk factors, which conclude $R_1$, $R_2$, $R_3$. The matrix unit is denoted as $G_{i,j}$, $W_k$, $R_l$, which indicates the possibility of the $R_l$ risk that occurs in the $G_{i,j}$ grid where the $W_k$ construction activity is located. For the $W_k$ construction activity, if the $R_l$ risk in the $G_{i,j}$ grid will occur, the matrix will be defined as "1", if not, it will be defined as "0".

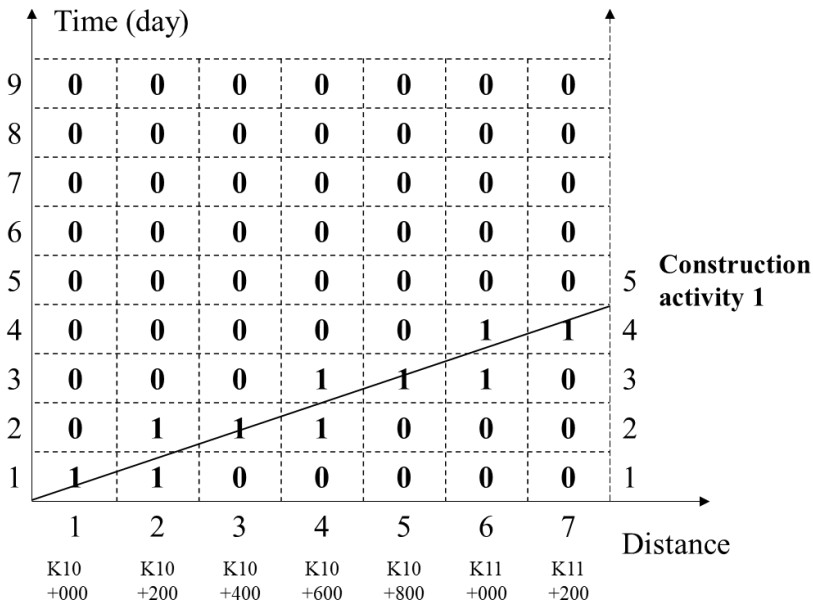

**Figure 4.** Railway construction schedule map and G–T–WBS matrix of construction activity 1.

**Table 1.** G–T–WBS–RBS matrix of construction activity 1.

| Risk Factor | Construction Activity 1 | | | | | | | | | |
|---|---|---|---|---|---|---|---|---|---|---|
| | $G_{1,1}$ | $G_{2,1}$ | $G_{2,2}$ | $G_{3,2}$ | $G_{4,2}$ | $G_{4,3}$ | $G_{5,3}$ | $G_{6,3}$ | $G_{6,4}$ | $G_{7,4}$ |
| $R_1$ | 1 | 1 | 1 | 1 | 0 | 0 | 0 | 0 | 1 | 1 |
| $R_2$ | 1 | 1 | 1 | 1 | 0 | 0 | 0 | 0 | 1 | 1 |
| $R_3$ | 1 | 1 | 1 | 1 | 0 | 0 | 0 | 0 | 1 | 1 |

The G–T–WBS–RBS matrix provides a visual representation of the link between the spatial–temporal attributes of construction activities and their potential risks. It can help users more intuitively discover the changes in a risk factor spatially and temporally, as well as the distribution and interaction of different risk factors spatially and temporally. Then, it can help users to dynamically analyze the generation and development of safety risks so as to formulate more effective risk control measures.

## 3. Case Study

In this paper, the A Railway was selected as a case study to demonstrate and validate the established safety risk identification model. The A Railway is located in the western of China and the construction environment of it is complex and dangerous. During the construction of the railway, it faced the influence of natural disasters such as strong winds, earthquakes, debris flow, and geological disasters such as rockburst and fault zones. The A Railway is characterized by huge engineering volumes, a complex construction environment, and harsh construction conditions. Based on the case study of the A Railway, it can effectively illustrate the scientificity and effectiveness of the safety risk identification model.

According to the "Railway Engineering Acceptance Standards Application Guide" for the division of construction engineering and the principle of decomposition level-by-level standards, combined with the actual situation of the A Railway construction, the A Railway

construction project is decomposed into civil engineering, power supply engineering, signal engineering, and track engineering. Among them, civil engineering can be divided into station construction, tunnel construction, bridge and culvert construction, and roadbed construction, as shown in Figure 5.

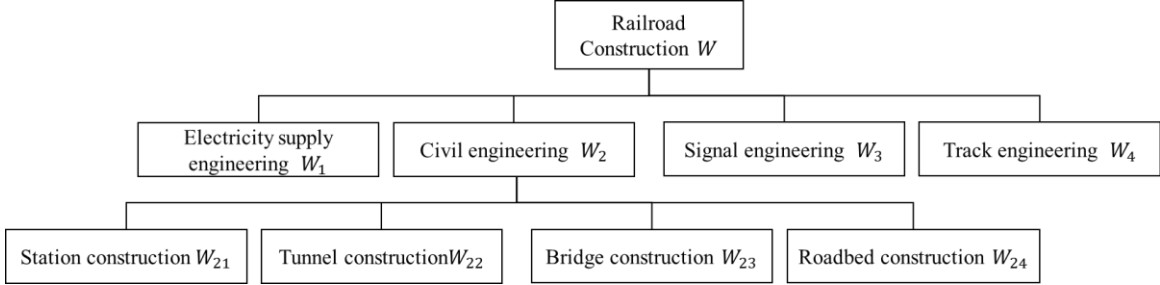

**Figure 5.** A Railway work breakdown structure tree.

For the safety risks of railway construction, Lu [33] divided them into four categories: climatic disasters, geologic hazards, hazards in plateaus, and engineering construction disasters. Dong [8] divides the construction risk of the China–Mongolia–Russia high-speed railway into three dimensions: economic, social, and ecological. Shen [34] classified the risks of high-speed railway construction into four dimensions: people, machine, management, and environment and identified nine risk factors, by using expert interview methods and literature analysis methods. Based on the above literature and the characteristics of the A railway, the RBS decomposition tree of the A Railway engineering is established in Table 2.

**Table 2.** The A Railway construction safety risk breakdown structure tree.

| Name | Tier 1 Risk Factors | Tier 2 Risk Factors |
|---|---|---|
| The A Railway construction safety risk $R$ | Climate Hazards $R_1$ | Rain and snow $R_{11}$ <br> Severe cold $R_{12}$ <br> Strong wind $R_{13}$ <br> Thunder and lightning $R_{14}$ |
| | Engineering Hazards $R_2$ | Temporary works collapse $R_{21}$ <br> Tunnel sudden water and mud $R_{22}$ <br> Bridge/tunnel collapse $R_{23}$ <br> Gas explosion $R_{24}$ <br> Scaffolding instability $R_{25}$ <br> Fall from height $R_{26}$ <br> Mold burst $R_{27}$ |
| | Geological Hazards $R_3$ | Mudslide $R_{31}$ <br> Avalanche $R_{32}$ <br> Landslide $R_{33}$ <br> Glacial eruption $R_{34}$ <br> Tunnel deformation $R_{35}$ <br> Dangerous rock fall $R_{36}$ <br> High Geothermal $R_{37}$ <br> Karst $R_{38}$ <br> High Temperature Water $R_{39}$ <br> Rock explosion $R_{310}$ <br> Seismic $R_{311}$ |

**Table 2.** *Cont.*

| Name | Tier 1 Risk Factors | Tier 2 Risk Factors |
|---|---|---|
| | Highland Hazards $R_4$ | Occupational disease $R_{41}$<br>Acute/chronic plateau disease $R_{42}$<br>Diseases of natural epidemic origin $R_{43}$<br>Other infectious diseases $R_{44}$<br>Accidental injury $R_{45}$<br>Plague $R_{46}$ |
| | Social Stability Risks $R_5$ | Customs and traditions $R_{51}$<br>Religious beliefs $R_{52}$<br>Road–land relations $R_{53}$ |

Based on the WBS tree and the RBS tree as well as construction information related to the A Railway, the G–T–WBS matrix of the A Railway construction project is constructed, and the results are shown in Table 3. In the process of establishing the risk management grid, the matrix form of each subproject was huge, and the risk management grid was complex and redundant. Due to the limitation of space, this paper only lists the risk identification grids corresponding to the representative civil engineering construction subprojects. Among them, $G_{1,13}$ indicates the construction content of the civil engineering project on the 13th day of construction within the starting and finishing mileage DK0 + 0.0~DK0 + 200.0, and $G_{11,16}$ indicates the construction content of the civil engineering project on the 16th day of construction within the starting and finishing mileage DK2 + 0.0~DK2 + 200.0.

**Table 3.** The G–T–WBS matrix of the A Railway civil engineering.

| Engineering Project | G-T-WBS Matrix |
|---|---|
| The A Railway civil engineering $W_2$ | $G_{1,13}$, $G_{2,13}$, $G_{3,13}$, $G_{4,13}$, $G_{4,14}$, $G_{5,14}$, $G_{5,14}$, $G_{5,14}$, $G_{6,15}$, $G_{7,15}$, $G_{8,15}$, $G_{9,15}$, $G_{10,15}$<br>$G_{11,16}$, $G_{12,16}$, $G_{13,16}$, $G_{14,16}$, $G_{14,17}$, $G_{15,17}$, $G_{16,17}$, $G_{17,17}$, $G_{17,18}$, $G_{18,18}$, $G_{19,18}$, $G_{20,18}$<br>$G_{21,19}$, $G_{22,19}$, $G_{23,19}$, $G_{24,19}$, $G_{24,20}$, $G_{25,20}$, $G_{26,20}$, $G_{27,20}$, $G_{27,21}$, $G_{28,21}$, $G_{29,21}$, $G_{30,21}$<br>$G_{31,22}$, $G_{32,22}$, $G_{33,22}$, $G_{34,22}$, $G_{34,23}$, $G_{35,23}$, $G_{36,23}$, $G_{37,23}$, $G_{37,24}$, $G_{38,24}$, $G_{39,24}$, $G_{40,24}$,<br>$G_{41,25}$, $G_{42,25}$, $G_{43,25}$, $G_{44,25}$, $G_{44,26}$, $G_{45,26}$, $G_{46,26}$, $G_{47,26}$, $G_{47,27}$, $G_{48,27}$, $G_{49,27}$, $G_{50,27}$<br>$G_{51,28}$, $G_{52,28}$, $G_{53,28}$, $G_{54,28}$, $G_{54,29}$, $G_{55,29}$, $G_{56,29}$, $G_{57,29}$, $G_{57,30}$, $G_{58,30}$, $G_{59,30}$, $G_{60,30}$<br>$G_{61,31}$, $G_{62,31}$, $G_{63,31}$, $G_{64,31}$, $G_{64,32}$, $G_{65,32}$, $G_{66,32}$, $G_{67,32}$, $G_{67,33}$, $G_{68,33}$, $G_{69,33}$, $G_{70,33}$ |

Based on the G–T–WBS matrix of the A Railway that has been constructed above, the G–T–WBS–RBS matrix shown in the Table 4 was established by inviting various experts, managers, designers, technicians, and construction-experienced workers of the A Railway to make judgments on the matrix, and considering the problem of space, only part of the risk matrix is listed in this paper.

**Table 4.** The G–T–WBS–RBS matrix of the A Railway civil engineering.

| Risk Factor | | $G_{10,30}$ $W_{21}$ | $G_{10,31}$ $W_{21}$ | $G_{10,32}$ $W_{21}$ | $G_{10,33}$ $W_{21}$ | $G_{40,30}$ $W_{22}$ | $G_{41,70}$ $W_{22}$ | $G_{42,110}$ $W_{22}$ | $G_{43,150}$ $W_{22}$ | $G_{91,40}$ $W_{23}$ | $G_{92,50}$ $W_{23}$ | $G_{93,60}$ $W_{23}$ | $G_{94,70}$ $W_{23}$ | $G_{65,35}$ $W_{24}$ | $G_{66,40}$ $W_{24}$ | $G_{67,45}$ $W_{24}$ | $G_{68,50}$ $W_{24}$ |
|---|---|---|---|---|---|---|---|---|---|---|---|---|---|---|---|---|---|
| | | | | | | | | | Civil Engineering $W_2$ | | | | | | | | |
| $R_1$ | $R_{11}$ | 1 | 1 | 1 | 1 | 0 | 0 | 0 | 0 | 1 | 1 | 0 | 1 | 0 | 1 | 0 | 1 |
| | $R_{12}$ | 1 | 1 | 1 | 1 | 0 | 0 | 0 | 0 | 1 | 1 | 1 | 1 | 1 | 1 | 1 | 1 |
| | $R_{13}$ | 1 | 1 | 1 | 1 | 0 | 0 | 0 | 0 | 1 | 1 | 0 | 1 | 0 | 1 | 0 | 1 |
| | $R_{14}$ | 1 | 0 | 0 | 0 | 0 | 0 | 0 | 0 | 0 | 1 | 0 | 0 | 0 | 0 | 0 | 1 |
| $R_2$ | $R_{21}$ | 1 | 1 | 1 | 1 | 1 | 1 | 1 | 1 | 1 | 1 | 0 | 1 | 0 | 1 | 0 | 1 |
| | $R_{22}$ | 0 | 0 | 0 | 0 | 1 | 1 | 1 | 1 | 0 | 0 | 0 | 0 | 0 | 0 | 0 | 0 |
| | $R_{23}$ | 0 | 0 | 0 | 0 | 1 | 1 | 1 | 1 | 1 | 1 | 1 | 1 | 0 | 0 | 0 | 0 |
| | $R_{24}$ | 0 | 0 | 0 | 0 | 0 | 1 | 0 | 0 | 0 | 0 | 0 | 0 | 0 | 0 | 0 | 0 |
| | $R_{25}$ | 1 | 1 | 1 | 1 | 1 | 0 | 0 | 1 | 1 | 1 | 0 | 1 | 0 | 0 | 0 | 0 |
| | $R_{26}$ | 1 | 1 | 1 | 1 | 0 | 0 | 0 | 0 | 1 | 1 | 0 | 1 | 0 | 0 | 0 | 0 |
| | $R_{27}$ | 0 | 0 | 0 | 0 | 1 | 1 | 1 | 1 | 1 | 0 | 1 | 0 | 0 | 0 | 0 | 0 |

**Table 4.** *Cont.*

| Risk Factor | | $G_{10,30}$ $W_{21}$ | $G_{10,31}$ $W_{21}$ | $G_{10,32}$ $W_{21}$ | $G_{10,33}$ $W_{21}$ | $G_{40,30}$ $W_{22}$ | $G_{41,70}$ $W_{22}$ | $G_{42,110}$ $W_{22}$ | $G_{43,150}$ $W_{22}$ | $G_{91,40}$ $W_{23}$ | $G_{92,50}$ $W_{23}$ | $G_{93,60}$ $W_{23}$ | $G_{94,70}$ $W_{23}$ | $G_{65,35}$ $W_{24}$ | $G_{66,40}$ $W_{24}$ | $G_{67,45}$ $W_{24}$ | $G_{68,50}$ $W_{24}$ |
|---|---|---|---|---|---|---|---|---|---|---|---|---|---|---|---|---|---|
| | | | | | | | | Civil Engineering $W_2$ | | | | | | | | | |
| | $R_{31}$ | 0 | 0 | 0 | 0 | 0 | 0 | 0 | 0 | 1 | 1 | 0 | 1 | 0 | 1 | 0 | 1 |
| | $R_{32}$ | 0 | 0 | 0 | 0 | 0 | 0 | 0 | 0 | 1 | 0 | 0 | 0 | 0 | 1 | 0 | 0 |
| | $R_{33}$ | 0 | 0 | 0 | 0 | 1 | 0 | 0 | 1 | 1 | 1 | 0 | 1 | 0 | 1 | 0 | 1 |
| | $R_{34}$ | 0 | 0 | 0 | 0 | 0 | 0 | 0 | 0 | 0 | 1 | 0 | 0 | 0 | 1 | 0 | 1 |
| | $R_{35}$ | 0 | 0 | 0 | 0 | 1 | 1 | 1 | 1 | 0 | 0 | 0 | 0 | 0 | 0 | 0 | 0 |
| $R_3$ | $R_{36}$ | 0 | 0 | 0 | 0 | 1 | 0 | 0 | 1 | 1 | 0 | 0 | 1 | 0 | 1 | 0 | 1 |
| | $R_{37}$ | 0 | 0 | 0 | 0 | 1 | 1 | 0 | 1 | 0 | 0 | 0 | 0 | 0 | 0 | 0 | 0 |
| | $R_{38}$ | 0 | 0 | 0 | 0 | 1 | 1 | 0 | 1 | 0 | 0 | 0 | 0 | 0 | 0 | 0 | 0 |
| | $R_{39}$ | 0 | 0 | 0 | 0 | 1 | 1 | 0 | 1 | 0 | 0 | 0 | 0 | 0 | 0 | 0 | 0 |
| | $R_{310}$ | 0 | 0 | 0 | 0 | 0 | 1 | 1 | 0 | 0 | 0 | 0 | 0 | 0 | 0 | 0 | 0 |
| | $R_{311}$ | 1 | 0 | 0 | 0 | 1 | 0 | 0 | 0 | 1 | 0 | 0 | 1 | 1 | 0 | 0 | 0 |
| | $R_{41}$ | 0 | 0 | 0 | 0 | 0 | 1 | 1 | 0 | 0 | 0 | 0 | 0 | 0 | 0 | 0 | 0 |
| | $R_{42}$ | 0 | 0 | 0 | 1 | 0 | 0 | 1 | 1 | 0 | 0 | 0 | 0 | 0 | 0 | 0 | 1 |
| $R_4$ | $R_{43}$ | 0 | 0 | 0 | 0 | 0 | 0 | 1 | 1 | 0 | 0 | 0 | 0 | 0 | 0 | 0 | 0 |
| | $R_{44}$ | 0 | 0 | 0 | 0 | 0 | 0 | 0 | 1 | 0 | 0 | 0 | 0 | 0 | 0 | 0 | 0 |
| | $R_{45}$ | 1 | 1 | 1 | 1 | 1 | 1 | 1 | 1 | 1 | 1 | 1 | 1 | 1 | 1 | 1 | 1 |
| | $R_{46}$ | 1 | 1 | 1 | 1 | 1 | 1 | 1 | 1 | 0 | 0 | 0 | 0 | 0 | 0 | 0 | 0 |
| | $R_{51}$ | 0 | 0 | 0 | 0 | 1 | 1 | 1 | 1 | 0 | 0 | 0 | 0 | 1 | 1 | 1 | 1 |
| $R_5$ | $R_{52}$ | 0 | 0 | 0 | 0 | 1 | 1 | 1 | 1 | 0 | 0 | 0 | 0 | 1 | 1 | 1 | 1 |
| | $R_{53}$ | 0 | 0 | 0 | 0 | 1 | 1 | 1 | 1 | 0 | 0 | 0 | 0 | 1 | 1 | 1 | 1 |

Other projects, by similarly constructing the G–T–WBS–RBS matrix, are able to identify the risks in the corresponding grid of a certain construction activity so as to obtain the risk identification results of railway project construction.

## 4. Discussions

The G–T–WBS–RBS matrix can reflect the distribution characteristics of risks in space and time and help researchers better analyze and study the interrelationships among risks and between risks and subprojects. For example, by directly observing the matrix, it can be found that the risk of climate hazards has less influence on the construction of tunnels, while the subprojects with more open-air construction situations, such as bridge construction, station construction, and roadbed construction, are more likely to receive the influence of climate hazards during the construction process. In addition, compared with roadbed and tunnel construction, bridge construction and station construction have more work-at-height tasks which are more prone to the risk of engineering hazards such as scaffolding instability and falls from height. As far as geological hazards are concerned, since stations are considered in the process of site selection, they are usually located on safer terrain and are seldom affected by geological hazards such as mudslides, avalanches, and landslides during the construction process. Bridge construction and roadbed construction, however, are more likely to be affected by mudslides and other geological hazards because they are directly exposed to the natural environment. Because the A Railway is located in western China, the geological activities along the railway line are intense, so the tunnel construction process is highly susceptible to geological hazards, such as tunnel deformation, high geothermal, karst, etc. Therefore, during the tunnel construction process, geological advance forecasting should be conducted to ensure the safety of tunnel construction. In addition, because the A Railway passes through the intersection of plates, seismic hazards along the line are frequent, and all subprojects need to do a good job in the construction process of seismic hazard prevention measures to reduce the impact of earthquakes on construction safety. In terms of highland hazards, occupational diseases are more likely to occur in the construction process of tunnel projects with harsh construction environments, while plague and infectious diseases are more likely to occur in the construction process of stations or tunnel projects with dense construction site personnel or relatively closed air. Plateau diseases and accidental injuries occur in the construction process of all kinds of projects, so it is necessary to develop corresponding prevention and treatment measures to ensure the safety of construction personnel. For the social stability risk, it is mainly concentrated in the subprojects that damage the local environment, such as tunnels and roadbeds, etc. During the construction preparation stage, we should do a good job of

propaganda and coordinate the relationship with the local people so as to obtain a good public foundation for the construction.

As for the relationship between the risks, the correlation can be analyzed by extracting the information related to each risk factor within the G–T–WBS–RBS matrix. For example, Table 5 shows the information of five types of risk factors in the matrix, such as rain, snow, strong wind, temporary works collapse, scaffolding instability, and falls from height; through observation, it can be found that the risks of rain, snow, strong wind, temporary works collapse, and scaffolding instability have a strong mutual relationship, which can be understood as that bad weather, such as rain, snow, and strong wind, will increase the probability of temporary works collapsing and scaffolding instabilities during the construction process Therefore, in the construction process, when encountering bad weather, working at height should be stopped and all kinds of temporary facilities and structures should be protected to prevent the occurrence of risk events.

**Table 5.** The $R_1$, $R_2$ parts of the G–T–WBS–RBS matrix.

| Risk Factor | | The A Railway Civil Engineering $W_2$ | | | | | | | | | | | | | | | |
|---|---|---|---|---|---|---|---|---|---|---|---|---|---|---|---|---|---|
| | | $G_{10,30}$ $W_{21}$ | $G_{10,31}$ $W_{21}$ | $G_{10,32}$ $W_{21}$ | $G_{10,33}$ $W_{21}$ | $G_{40,30}$ $W_{22}$ | $G_{41,70}$ $W_{22}$ | $G_{42,110}$ $W_{22}$ | $G_{43,150}$ $W_{22}$ | $G_{91,40}$ $W_{23}$ | $G_{92,50}$ $W_{23}$ | $G_{93,60}$ $W_{23}$ | $G_{94,70}$ $W_{23}$ | $G_{65,35}$ $W_{24}$ | $G_{66,40}$ $W_{24}$ | $G_{67,45}$ $W_{24}$ | $G_{68,50}$ $W_{24}$ |
| $R_1$ | $R_{11}$ | 1 | 1 | 1 | 1 | 0 | 0 | 0 | 0 | 1 | 1 | 0 | 1 | 0 | 1 | 0 | 1 |
| | $R_{13}$ | 1 | 1 | 1 | 1 | 0 | 0 | 0 | 0 | 1 | 1 | 0 | 1 | 0 | 1 | 0 | 1 |
| | $R_{21}$ | 1 | 1 | 1 | 1 | 1 | 1 | 1 | 1 | 1 | 1 | 0 | 1 | 0 | 1 | 0 | 1 |
| $R_2$ | $R_{25}$ | 1 | 1 | 1 | 1 | 1 | 0 | 0 | 1 | 1 | 1 | 0 | 1 | 0 | 0 | 0 | 0 |
| | $R_{26}$ | 1 | 1 | 1 | 1 | 0 | 0 | 0 | 0 | 1 | 1 | 0 | 1 | 0 | 0 | 0 | 0 |

In addition, by extracting the information related to the risks of rain, snow, strong winds, mudslides, avalanches, landslides, and dangerous rock falls, Table 6 can be formed. By observing the table, we can find that the risks of rain, snow, and strong winds, and mudslides, landslides, and dangerous rock falls also have a strong mutual relationship, which is likely because the strong rainfall generated by rain and snow can easily induce the occurrence of mudslides and other disasters, and the long-term erosion of mountains by strong winds is also the main cause of dangerous rock-fall disasters.

**Table 6.** The $R_1$, $R_3$ parts of the G–T–WBS–RBS matrix.

| Risk Factor | | The A Railway Civil Engineering $W_2$ | | | | | | | | | | | | | | | |
|---|---|---|---|---|---|---|---|---|---|---|---|---|---|---|---|---|---|
| | | $G_{10,30}$ $W_{21}$ | $G_{10,31}$ $W_{21}$ | $G_{10,32}$ $W_{21}$ | $G_{10,33}$ $W_{21}$ | $G_{40,30}$ $W_{22}$ | $G_{41,70}$ $W_{22}$ | $G_{42,110}$ $W_{22}$ | $G_{43,150}$ $W_{22}$ | $G_{91,40}$ $W_{23}$ | $G_{92,50}$ $W_{23}$ | $G_{93,60}$ $W_{23}$ | $G_{94,70}$ $W_{23}$ | $G_{65,35}$ $W_{24}$ | $G_{66,40}$ $W_{24}$ | $G_{67,45}$ $W_{24}$ | $G_{68,50}$ $W_{24}$ |
| $R_1$ | $R_{11}$ | 1 | 1 | 1 | 1 | 0 | 0 | 0 | 0 | 1 | 1 | 0 | 1 | 0 | 1 | 0 | 1 |
| | $R_{13}$ | 1 | 1 | 1 | 1 | 0 | 0 | 0 | 0 | 1 | 1 | 0 | 1 | 0 | 1 | 0 | 1 |
| | $R_{31}$ | 0 | 0 | 0 | 0 | 0 | 0 | 0 | 0 | 1 | 1 | 0 | 1 | 0 | 1 | 0 | 1 |
| | $R_{32}$ | 0 | 0 | 0 | 0 | 0 | 0 | 0 | 0 | 1 | 0 | 0 | 0 | 0 | 1 | 0 | 0 |
| $R_3$ | $R_{33}$ | 0 | 0 | 0 | 0 | 1 | 0 | 0 | 1 | 1 | 1 | 0 | 1 | 0 | 1 | 0 | 1 |
| | $R_{34}$ | 0 | 0 | 0 | 0 | 0 | 0 | 0 | 0 | 0 | 1 | 0 | 0 | 0 | 1 | 0 | 1 |
| | $R_{36}$ | 0 | 0 | 0 | 0 | 1 | 0 | 0 | 1 | 1 | 0 | 0 | 1 | 0 | 1 | 0 | 1 |

Based on the above analysis, it can be found that by using the risk identification method based on the G–T–WBS–RBS matrix, the spatial and temporal distribution of the risk factors and their interactions can be found more intuitively. Some risk factors that are not closely related subjectively will unexpectedly show strong correlations in the matrix. Given that only part of the railway project was analyzed in this case study, more information would be found if the scope of application of the risk identification method was increased.

Through the G–T–WBS–RBS matrix, the relationship between risk factors and risk engineering can be reflected more intuitively to facilitate researchers or construction personnel to identify and analyze risks, so that in the construction preparation stage, the corresponding risk prevention, control, and emergency measures are targeted to ensure the maximum construction safety of railway projects, laying the foundation for the smooth implementation of railway construction.

## 5. Conclusions

This paper constructs a comprehensive method applicable to the safety risk identification of railway construction in complex and dangerous areas. The method completes risk identification by using the WBS and the RBS to analyze railway projects and form a G–T–WBS–RBS matrix based on the interrelationship between construction activities and their risk factors. In this paper, the A Railway located in western China was selected for the case study, and by constructing the G–T–WBS–RBS matrix of the A Railway and analyzing it, the following conclusions were obtained:

1.  During the construction of the A Railway, bridge construction, station construction, and roadbed construction are most likely to receive climate hazards, while tunnel construction is more likely to be affected by geological hazards. Because of the intense geological activities along the railway line, it is necessary to take precautionary measures against seismic hazards during the construction process. Diseases represented by plague and occupational diseases are prone to occur during the construction of stations or tunnels where the construction site is crowded, or the air is closed. Based on the interrelation between subprojects and their risks, project managers can develop targeted risk prevention measures during the construction process. For climate hazards, managers can build perfect climate monitoring and early warning means and take effective risk control measures before the emergence of extreme climate. For geological hazards, managers should do a good job of geological prediction in the construction process of tunnel engineering and predict the possible geological disasters in time. For diseases, managers should always keep the environment of the construction site healthy to avoid the generation and transmission of diseases.

2.  Risk events are generated by the coupling of risk factors, so the interrelationship of risk factors can reflect the evolutionary process of risk to some extent. By analyzing the interrelationship between the safety risks of the A Railway civil engineering, it can be found that rain, snow, strong winds, and scaffolding instability always happen in the same space and time, and they have a strong mutual relationship, which indicates that bad weather is an important cause of scaffolding instabilities. Similarly, rain, snow, mudslides, and dangerous rock falls have the same interrelationship. Based on the results of this study, managers can strengthen scaffolding and support structures before extreme weather events occur. In addition, it is also necessary to organize the construction personnel to avoid the natural hazards caused by the extreme weather in time. By focusing on the distribution of risk factors in space and time and finding risk factors with strong interrelationships, the analysis results can provide an important theoretical basis for risk evolution analysis.

A comprehensive identification of risk factors can be achieved by using this risk identification method to analyze the risk components of railway projects and their spatial and temporal distribution characteristics, which provides a basis for analyzing the characteristics of risk factors and the evolution of risk processes. However, this method still has some limitations. It is difficult to construct a complete G–T–WBS–RBS matrix for the total railway project considering the large volume of work and long construction cycle. The construction of the matrix will take a lot of time and needs to collect a lot of data related to the railway project. Taking these factors into account, subsequent research can address this difficulty and explore how to use this method for efficient risk identification based on computer technology. In addition, the resulting G–T–WBS–RBS matrix will contain a large amount of data, and some scientific data analysis methods can be considered when analyzing it.

**Author Contributions:** Methodology, G.Z.; Project administration, Q.W.; Supervision, J.W.; Writing–original draft, P.W.; Writing–review & editing, Y.Y. All authors have read and agreed to the published version of the manuscript.

**Funding:** This research was funded by the National Natural Science Foundation of China grant number 71941014.

**Data Availability Statement:** All datasets generated for this study are included in this paper.

**Conflicts of Interest:** The authors declare no conflict of interest.

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
