# Peer review of "Safety Risk Identification Method for Railway Construction in Complex and Dangerous Areas"

_sustainability, doi:10.3390/su142113698_

Round 1
Reviewer 1 Report
In the abstract, it is not straightforward to see what the CZ railway is.
The abbreviations like WBS, and RBS should be explained at their first appearance.
In the introduction, it is recommended to indicate that the complex and dangerous areas not only affect the operation of the railway system [1-2], but also pose challenges to the construction of the rail infrastructure.
[1] Wind deflection analysis of railway catenary under crosswind based on nonlinear finite element model and wind tunnel test. Mechanism and Machine Theory, 2022. 168, 104608.
[2] "Rail vehicle design optimisation for operation in a mountainous railway track." Innovative Infrastructure Solutions 2.1 (2017): 1-6.
In the last paragraph of the introduction, it is likely that the novelty of this paper is to consider the risk characteristics. Could the authors give more explanations on this point?
A more specific title is recommended for Section 2. 'Materials and Methods' is too general.
It is not straightforward to see what is the main novelty of the method used or the analysis conducted in this paper against previous research. Please notice that a simple application of a mature method to an object (CZ railway in this paper) is difficult to be claimed as a novelty. Please comment on this issue.
A reference should be mentioned for the 'linear plan diagram' as this method is not proposed in this paper firstly.
Why 4,3 is cycled in figure 2? It looks that the red cycle is not explained.
The information on the CZ railway is not given. The description in Section 3 is very general, and it is hard to see the 'CASE study'.
Generally, the reviewer appreciates this work to implement a safety risks identification for railway construction in complex and dangerous. But the biggest problem is that the case study description is too simple, and the readers cannot see why the CZ railway analysed in this paper is complex and dangerous. More discussion should be given.
This paper can be published after the above issues are addressed.
Author Response
Response to Reviewer 1
Dear reviewer,
We would like to thank you for your careful reading, helpful comments, and constructive suggestions, which has significantly improved the presentation of our manuscript.
We have carefully considered all comments from you and revised our manuscript accordingly. In the following section, we summarize our responses to each comment from you.
- In the abstract, it is not straightforward to see what the CZ railway is.
Response: We have added the introduction of the CZ railway in the abstract so that readers can have a clearer understanding of it.
- The abbreviations like WBS, and RBS should be explained at their first appearance.
Response: The abbreviations have been explained at their first appearance.
- In the introduction, it is recommended to indicate that the complex and dangerous areas not only affect the operation of the railway system [1-2], but also pose challenges to the construction of the rail infrastructure.
[1] Wind deflection analysis of railway catenary under crosswind based on nonlinear finite element model and wind tunnel test. Mechanism and Machine Theory, 2022. 168, 104608.
[2] "Rail vehicle design optimisation for operation in a mountainous railway track." Innovative Infrastructure Solutions 2.1 (2017): 1-6.
Response: Now, the affections and challenges caused by the complex and dangerous areas have been indicated.
- In the last paragraph of the introduction, it is likely that the novelty of this paper is to consider the risk characteristics. Could the authors give more explanations on this point?
Response: We given more explanations for the novelty of this paper in the last paragraph of the introduction, and we have introduced the characteristics of risk in detail.
- A more specific title is recommended for Section 2. 'Materials and Methods' is too general.
Response: Now the title of Section 2 has been changed to “The Safety Risks Identification Method Based on The G-T-WBS-RBS Matrix”, which better reflects the research content of Section 2.
- It is not straightforward to see what is the main novelty of the method used or the analysis conducted in this paper against previous research. Please notice that a simple application of a mature method to an object (CZ railway in this paper) is difficult to be claimed as a novelty. Please comment on this issue.
Response: This paper constructs a new risk identification method based on G-T-WBS-RBS matrix. This risk identification method can help users more intuitively discover the distribution and interaction of risk factors, so as to help them formulate more effective risk control measures. In addition, this paper is based on the CZ Railway, a large railway construction in a complex and dangerous area, to conduct a case study. The results show that this method can help users to understand the generation and development of safety risks more clearly.
- A reference should be mentioned for the 'linear plan diagram' as this method is not proposed in this paper firstly.
Response: There was a presentation error here, which we have corrected.
- Why 4,3 is cycled in figure 2? It looks that the red cycle is not explained.
Response: Now, it is explained in the second paragraph of section 2.3
- The information on the CZ railway is not given. The description in Section 3 is very general, and it is hard to see the 'CASE study'.
Response: More information of the CZ Railway has been added in Section 3.
- Generally, the reviewer appreciates this work to implement a safety risks identification for railway construction in complex and dangerous. But the biggest problem is that the case study description is too simple, and the readers cannot see why the CZ railway analysed in this paper is complex and dangerous. More discussion should be given.
Response: Now, we have given a more specific description of the case study, so as to help readers more clearly understand the complex and dangerous environment of CZ railway construction.
Thank you for your careful reading again.

Reviewer 2 Report
1. Abstract - Please provide some quantitative findings in the abstract. Also, highlight the key contributions and applications of the study in the end.
2. Title - How do the authors differentiate between complex and dangerous? What do these terms indicate? A more precise and self-explanatory terminology would be better.
3. At the end of introduction, the authors can make a subsection on research motivation and objectives where they can highlight the research need and contributions to the state of art. At present, the novelty is not clear.
4. The authors need to discuss their findings in light of the past research. For example, what are the similarities and contradictions of the results in comparison with other case studies from outside China.
5. There are no limitations of the study mentioned in the paper. No study is complete without reporting its limitations.
6. In the conclusions, the authors need to discuss how the study findings can be useful for policymakers and traffic engineering officials.
Author Response
Response to Reviewer 2
Dear reviewer,
We would like to thank you for your careful reading, helpful comments, and constructive suggestions, which has significantly improved the presentation of our manuscript.
We have carefully considered all comments from you and revised our manuscript accordingly. In the following section, we summarize our responses to each comment from you.
- Abstract - Please provide some quantitative findings in the abstract. Also, highlight the key contributions and applications of the study in the end.
Response: The relevant content has been supplemented in the abstract ‘to highlight the key contributions and applications of the study.
- Title - How do the authors differentiate between complex and dangerous? What do these terms indicate? A more precise and self-explanatory terminology would be better.
Response: The definition of complex and dangerous have been added in the first paragraph of the introduction.
- At the end of introduction, the authors can make a subsection on research motivation and objectives where they can highlight the research need and contributions to the state of art. At present, the novelty is not clear.
Response: The end of introduction has been corrected according to the comment.
- The authors need to discuss their findings in light of the past research. For example, what are the similarities and contradictions of the results in comparison with other case studies from outside China.
Response: In this paper, the results of this paper have been discussed in more detail based on previous studies.
- There are no limitations of the study mentioned in the paper. No study is complete without reporting its limitations.
Response: We have added a description of the limitations of this study at the end of the paper.
- In the conclusions, the authors need to discuss how the study findings can be useful for policymakers and traffic engineering officials.
Response: In the conclusion of the article, we give more policy suggestions based on the research results to help readers understand the practicability of this risks identification method.
Thank you for your careful reading again.

Reviewer 3 Report
1. Language problems exist. There are many long sentences, long paragraphs, grammatical errors, and phrases with ambiguous meanings, e.g., lines 13 to 17, etc. Words need to be checked throughout the whole manuscript, e.g., line 71(security risks), line 108(tress), etc. Moreover, abbreviations are suggested in entire illustrations when they first appear, e.g., WBS, RBS.
2. In general, the introduction and literature review is a bit on the light side. The authors need to clarify deeply their innovation and contribution in relation to the available literature. Especially, literature comparison on WBS-RBS for risk identification is suggested.
3. Only a framework of grid-time-WBS-RBS is proposed. The innovation and contribution seem a bit small to the current research. Moreover, the WBS-RBS is a traditional approach for risk identification. The advantage of combining gird-time is not fully discussed.
Author Response
Response to Reviewer 3
Dear reviewer,
We would like to thank you for your careful reading, helpful comments, and constructive suggestions, which has significantly improved the presentation of our manuscript.
We have carefully considered all comments from you and revised our manuscript accordingly. In the following section, we summarize our responses to each comment from you.
- Language problems exist. There are many long sentences, long paragraphs, grammatical errors, and phrases with ambiguous meanings, e.g., lines 13 to 17, etc. Words need to be checked throughout the whole manuscript, e.g., line 71(security risks), line 108(tress), etc. Moreover, abbreviations are suggested in entire illustrations when they first appear, e.g., WBS, RBS.
Response: We are very sorry for the trouble caused to your reading due to the language problems. These problems have been corrected in this paper.
- In general, the introduction and literature review is a bit on the light side. The authors need to clarify deeply their innovation and contribution in relation to the available literature. Especially, literature comparison on WBS-RBS for risk identification is suggested.
Response: We carefully revised the introduction section, and explained the innovation and contribution of this paper in more detail. Meanwhile, the research related to WBS-RBS was also added.
- Only a framework of grid-time-WBS-RBS is proposed. The innovation and contribution seem a bit small to the current research. Moreover, the WBS-RBS is a traditional approach for risk identification. The advantage of combining gird-time is not fully discussed.
Response: Based on the research content, the innovation of the framework of grid-time-WBS-RBS and the advantages of combining WBS-RBS with grid-time are explained in more detail in this paper.
Thank you for your careful reading again.

Round 2
Reviewer 1 Report
All my comments have been well addressed.
Reviewer 2 Report
The authors have significantly enhanced their manuscript based on the reviewers' suggestions. I accept this manuscript with a minor suggestion to get the entire manuscript proofread by a native English speaker or an English editing service to improve the use of language and grammar. Other than that, I have no further comments and the authors are appreciated for their novel work. Best wishes!
Reviewer 3 Report
Thanks for the modification and the great effort you have made.